# Pharmacokinetics and safety after once and twice a day doses of meclizine hydrochloride administered to children with achondroplasia

Hiroshi Kitoh[1,2]*, Masaki Matsushita[2], Kenichi Mishima[2], Tadashi Nagata[2], Yasunari Kamiya[2], Kohei Ueda[3], Yachiyo Kuwatsuka[3], Hiroshi Morikawa[4], Yasuhiro Nakai[3], Naoki Ishiguro[2]

1 Department of Orthopaedic Surgery, Aichi Children's Health and Medical Center, Obu, Aichi, Japan, 2 Department of Orthopaedic Surgery, Nagoya University Graduate School of Medicine, Showa-ku, Nagoya, Aichi, Japan, 3 Department of Advanced Medicine, Nagoya University Hospital, Showa-ku, Nagoya, Aichi, Japan, 4 Ina Research Inc, Ina, Nagano, Japan

* hkitoh420@gmail.com

**Data Availability Statement:** All relevant data are within the manuscript and its Supporting Information files.

## Abstract

Achondroplasia (ACH) is the most common short-limbed skeletal dysplasia caused by activating mutations in the fibroblast growth factor receptor 3 (*FGFR3*) gene. We identified that meclizine hydrochloride inhibited *FGFR3* signaling in various chondrocytic cells and promoted longitudinal bone growth in mouse model of ACH. Meclizine has safely been used for more than 50 years, but it lacks the safety data for repeated administration and pharmacokinetics (PK) when administered to children. We performed a phase Ia study to evaluate the PK and safety of meclizine administered orally to ACH children. Twelve ACH children aged from 5 to younger than 11 years were recruited, and the first 6 subjects received once a day of meclizine in the fasted condition, subsequent 6 subjects received twice a day of meclizine in the fed condition. Meclizine was well tolerated in ACH children with no serious adverse events. The mean $C_{max}$, $T_{max}$, $AUC_{0-24h}$, t1/2 during 24 hours in the fasted condition were 130 ng/mL, 1.7 hours, 761 ng·h/mL, and 8.5 hours respectively. The simulation of repeated administration of meclizine for 14 days demonstrated that plasma concentration apparently reached steady state around 10 days after the first dose both at once a day and twice a day administration. The $AUC_{0-10h}$ of the fasting and fed condition were 504 ng·h/mL and 813 ng·h/mL, respectively, indicating exposure of meclizine increased with the diet. Although higher drug exposure was confirmed in ACH children compared to adults, a single administration of meclizine seemed to be well tolerated.

## Introduction

Achondroplasia (ACH) is one of the most common skeletal dysplasias characterized by severe short stature with rhizomelic shortening of the extremities, relative macrocephaly with frontal bossing, midface hypoplasia, and increased lumbar lordosis. ACH is caused by activating mutations in *FGFR3* encoding the fibroblast growth factor receptor 3, which is a negative

**Funding:** HK received Advanced Medical Development Funds of Nagoya University Hospital and Grants from Japan Agency for Medical Research and Development. Hiroshi Morikawa is an employee of a commercial company (Ina Research Inc.), but the company provided support in the form of salaries for HM, and did not have any additional role in the study design, data collection and analysis, decision to publish, or preparation of the manuscript.

**Competing interests:** Hiroshi Morikawa is an employee of a commercial company (Ina Research Inc.), but the company provided support in the form of salaries for HM, and did not have any additional role in the study design, data collection and analysis, decision to publish, or preparation of the manuscript. This commercial affiliation does not alter our adherence to PLOS ONE policies on sharing data and materials.

regulator of longitudinal bone growth [1, 2]. In addition to severe short stature, neurological complications associated with stenosis of the foramen magnum and spinal canal, such as gait disturbance, leg paralysis, hydrocephalus, and central hypopnea, exacerbate patients' quality of life [3].

Growth hormone has been administered to ACH children for the treatment of short stature in some limited countries, but the response to this therapy is moderate [4]. Limb lengthening surgery still remains to be the most effective measure to increase height in ACH patients, but it involves significant period and effort [5]. Inhibition of *FGFR3* signaling is a therapeutic strategy for ACH, but no effective treatments are currently available. Several *FGFR3*-targeted treatments for ACH have experimentally been applied in recent years, including C-type natriuretic peptide (CNP) [6], NVP-BGJ398 (specific inhibitor for FGFR3) [7], anti-FGF2 aptamer [8], soluble FGFR3 (decoy receptor for FGFR3) [9], and statins [10]. The CNP analog, BMN111, is now undergoing a human clinical trial [11].

By comprehensive drug screening, we identified that meclizine hydrochloride (meclizine), an over-the-counter (OTC) histamine H1 receptor inhibitor used to treat motion sickness, inhibited *FGFR3* signaling in various chondrocytic cells [12]. We also confirmed that oral administration of clinically feasible dose of meclizine for the indication of "motion sickness" promoted longitudinal bone growth in mouse model of ACH [13, 14]. Meclizine is a prescription medicine approved in 1957 in the United States, and has been used as both prescription medicine and various OTC medicines in the world. Usage of OTC medicine containing meclizine of 25 mg a day for children over 3 years old, and 50 mg a day for children over 11 years old has been approved in Japan in 1988. But it lacks the data based on the current regulatory requirements including safety for repeated administrations. Following the advice of the Pharmaceuticals and Medical Devices Agency of Japan (PMDA), preclinical safety studies that equivalent to the development of new chemical entities along ICH M3 (R2) and ICH-E11, have been conducted. Prior to this phase 1a study, single dose TK studies of rats and dogs, 1 week and 2 weeks repeated dose toxicity studies of rats and dogs were completed in a contract laboratory (LSI Medicine Corp., Kashima, Japan). Currently, juvenile animal experiments are underway. There are, however, no pharmacokinetic studies of meclizine administered to children. According to the advice from PMDA, phase 1a study of not only 25 mg a day but also 50 mg a day for ACH children between 3 years old and 11years old, was planned to analyze the PK and safety of once and twice a day oral doses of meclizine. This study was conducted in the process of obtaining drug approval for the treatment of short stature in ACH.

## Materials and methods

This was a phase Ia, open-label study to evaluate the PK and safety of meclizine in two groups, the first one to be conducted as single administration, the second as twice a day administration. This study was conducted in Nagoya University Hospital, Nagoya, Japan under the Japanese Pharmaceuticals Affaires Low and GCP defined by the Ministry of Health, Labour and Welfare (MHLW) of Japan. Meclizine tablets were purchased from the OTC manufacturer (Meiji-yakuhin, Toyama, Japan) and verified as test drugs in study site. The study protocol was prepared after the consultations with the PMDA, and submitted to the PMDA together with the results of the preclinical studies. The study protocol and all amendments were reviewed and approved by the Institutional Review Boards of Nagoya University Hospital. This study is registered on University hospital Medical Information Network-Clinical Trials Registry (UMIN-CTR: ID:UMIN000033052) (https://upload.umin.ac.jp/cgi-open-bin/ctr_e/ctr_view.cgi?recptno=R000037683). All subjects or their legally authorized representative were required to provide written informed consent prior to participation in the study.

Subjects aged from 5 to younger than 11 years old who were diagnosed as ACH clinically (based on the Japanese diagnostic criteria of ACH) or genetically (*FGFR3* mutations) were eligible for enrollment. Key exclusion criteria included previous exposure to meclizine within 28 days prior to this study, surgical treatment for limb lengthening within 28 days, weight less than 11 kg, serious complications such as neurological impairments, clinically significant dysuria, history of glaucoma, and known hypersensitivity or allergies to meclizine.

The study was consisted of a screening period (Day -28 to Day -1), a treatment period (Day 1), and a follow-up period (Day 2 to Day 8). Screening procedures included obtaining the past medical history and physical examination, 12-lead electrocardiography (ECG), chest radiography, and assessment of laboratory parameters including haematology, blood chemistry, and urinalyses. Each subject was hospitalized in Nagoya University Hospital on Day -1, discharged on Day 2, and then visited the study site on Day 8. We determined two different doses of meclizine for the PK analysis according to our previous preclinical experiments using mouse model of ACH [14]. Since this was an exploratory study, it was not subject to a formal sample size calculation. However, based on similar pharmacokinetic studies, the sample size of 6 subjects per group was considered sufficient to meet the study objectives. Subjects were thus divided into two groups, the first 6 subjects were assigned to "once a day administration group", and subsequent 6 subjects were assigned to "twice a day administration group". On Day 1, subjects of "once a day group" received a single oral dose of meclizine 25 mg, in the form of a single tablet, with 150 mL of water after having fasted for > 10 hours, and continued to fast for additional one hour. On the other hand, subjects of "twice a day group" took the same dose of meclizine one hour after eating breakfast, and fasted for 4 hours after administration. They received 25 mg of meclizine again one hour after eating supper (10 hours after the first administration). Non-steroidal-anti-inflammatory drugs (NSAIDs) and various medicines containing anti-histamine were prohibited from 24 hours prior to treatment to 24 hours after administration. Anti-motion sickness drugs containing meclizine and reported drugs with an action to suppress *FGFR3* (CNP analog or statins) were also prohibited from 7 days prior to treatment to the end of the study (Day 8). Subjects were given consistent instructions regarding dietary intake time, administration time of the drug, and drugs prohibited from concomitant use throughout the study period and instructed to comply with them.

For the PK assessments, blood samples (2 mL per sample) were collected immediately before administration of the drug, and at 1, 2, 3, 4, 8, 10, and 24 hour after administration for "once a day group". For "twice a day group", samples were collected at 1, 2, 3, 4 hour after the second administration in addition to the above 8 time points. Catheters were used to avoid repeated use of a needle. Plasma concentration of meclizine was also measured at Day 8 for evaluation of drug accumulation in both groups. All blood samples were collected in tubes containing heparin sodium as the anticoagulant. Plasma was separated by centrifugation for 10 minutes at 3,000 rpm at room temperature. Afterward, the separated plasma samples were transferred into 2 polypropylene tubes and stored at approximately -40°C until analysis. Plasma concentration of meclizine was determined using validated liquid chromatography-tandem mass spectrometry (LC-MS/MS) assays at Ina Research Inc., Nagano, Japan. Meclizine in plasma samples was extracted by the protein precipitation method using methanol with flunarizine hydrochloride as the internal standard. Reverse-phase chromatography, employing a C18 column, a 10 mmol/L ammonium acetate aqueous solution containing 0.2% formic acid, and acetonitrile as the mobile phase, was used to determine meclizine concentrations by LC-MS/MS. The calibration curve was linear over the range 0.5–200 ng/mL. PK parameters were calculated for each subject with non-compartmental analysis of plasma concentration-time data: maximum concentration in plasma ($C_{max}$), time to reach $C_{max}$ ($T_{max}$), terminal elimination half-life (t1/2), and area under the plasma concentration-time curve from 0 to 24

hours post-dose ($AUC_{0-24}$) and from 0 to infinity ($AUC_{0-inf}$) [15]. PK parameters were evaluated using a non-compartmental model of Phoenix WinNonlin version 6.1 software (Pharsight, Mountain View, CA, USA).

To estimate the extent of drug accumulation after multiple dosing, we simulated changes in plasma concentration of meclizine when repeatedly administered once a day or twice a day for 14 days using the elimination rate constant (kel) determined based on the mean and the individual measured results after once or twice daily administration of meclizine. The kel was calculated using the terminal phase during 24 hours and 7 days after the first dose, respectively. The superposition method was adapted to simulate multiple administrations using first day's plasma concentration profile, plasma concentration at 24 hours, and kel.

For safety assessments, data regarding adverse events (AEs) based on physical examinations, vital signs (body temperature, blood pressure, and pulse rate), routine haematology, serum biochemistry, and urinalyses were collected at screening, throughout the hospitalized period (Day -1 to Day 2), and at Day 8 in both groups. Twelve-lead ECG was performed at screening, 4 hours after the first administration of meclizine, and at Day 8 in both groups. Safety data were summarized descriptively and presented in tabular format. All AEs reported by subjects or detected in the assessment were recorded, and the investigators determined their relationship to the treatment. The terminology and severity of the AEs were determined based on the Common Terminology Criteria for Adverse Events (CTCAE v4.03/MedDRA v12.0). Safety findings were summarized using descriptive statistics or frequency distributions.

All subjects who received at least one dose of meclizine were included in the safety analysis. Incidence and 95% confidence of all AEs were calculated according to the Clopper-Pearson exact method. All subjects who completed all measurements of plasma concentration of meclizine were included in the PK analysis. The primary evaluation points were PK parameters including $C_{max}$, $T_{max}$, t1/2, and AUC, and accumulation of meclizine 7 days after administration. AEs, vital signs, 12-lead ECG findings, and laboratory results served as safety evaluation points. The secondary evaluation point was simulated PK after repeated dose.

## Results

A single institution, phase Ia, open-label, once and twice a day doses study in ACH children was conducted between July 2018 and November 2018. A total of 12 ACH children (7 males and 5 females) signed the informed consent and were enrolled in the study (Fig 1). Early 6 subjects (MEC-01 to MEC-06) received 25 mg of meclizine once a day and subsequent 6 subjects (MEC-07 to MEC-12) had the same dose of meclizine twice a day. Baseline characteristics of the subjects were shown in Table 1. The median age of subjects was 8 years (interquartile range [IQR], 6–9 years) in "once a day group" and 7 years (IQR, 5–10 years) in "twice a day group". The median weight and height were 20.8 kg (IQR, 17.5–24.1 kg) and 102.0 cm (IQR, 95.5–109.0 cm) in "once a day group" and 21.2 kg (IQR, 15.3–30.1 kg) and 103.4 cm (IQR, 83.1–116.0 cm) in "twice a day group", respectively. The body mass index-standard deviation score (BMI-SDS) ranged from 0.81 to 2.40 in "once a day group" and from 1.40 to 3.36 in "twice a day group". The median body surface area (BSA), calculated according to the DuBois methods, was 0.735 $m^2$ (IQR, 0.661–0.833 $m^2$) in "once a day group" and 0.759 $m^2$ (IQR, 0.552–0.964 $m^2$) in "twice a day group". All but one subject (MEC-9) showed Tanner stage 1 corresponded to the pre-pubertal. All subjects continued growth hormone therapy during the study period that had been undertaken before the study. Eleven subjects had not received limb lengthening surgery. One subject (MEC-09) aged 10 years old had underwent bilateral tibial lengthening surgery 2 months prior to meclizine treatment, and she was completed throughout the study during hospitalization.

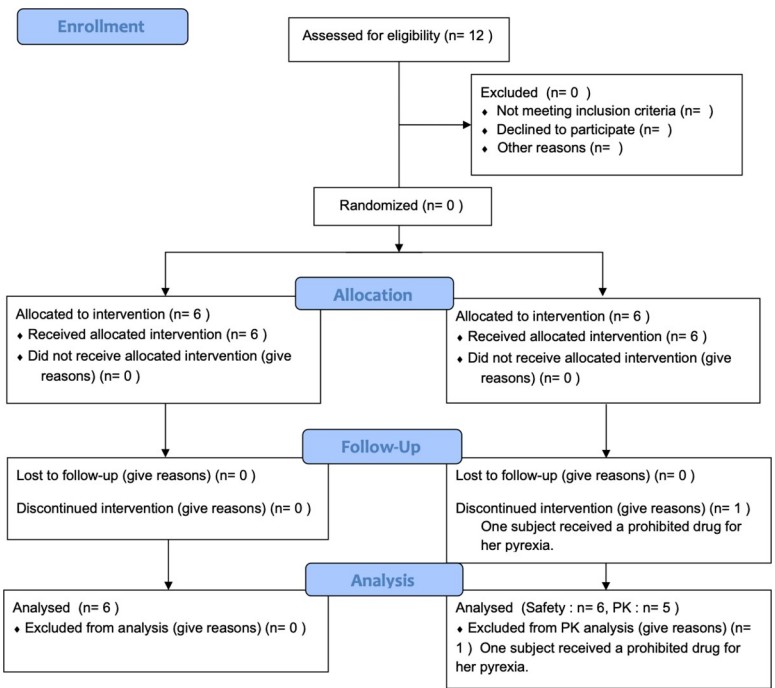

**Fig 1. CONSORT flowchart.**

All subjects were included in the safety analyses. There were no findings of clinical concern from the laboratory tests, vital signs, physical examinations, and ECG assessments. The AEs occurred in 3 subjects (5 events): subject MEC-05 developed somnolence, subject MEC-06

**Table 1. Summary of subject baseline characteristics.**

| Once a day administration of meclizine | | | | | | | | | |
|---|---|---|---|---|---|---|---|---|---|
| ID | Age (years) | Gender | Height (cm) | SD[a] for height | Weight (kg) | BMI[b] (kg/m2) | SD[a] for BMI | BSA[c] (m²) | Tanner Staging |
| MEC-01 | 6 | Male | 95.5 | -4.36 | 17.5 | 19.2 | 1.67 | 0.661 | 1 |
| MEC-02 | 10 | Male | 123.2 | -2.78 | 37.2 | 24.5 | 1.74 | 1.095 | 1 |
| MEC-03 | 8 | Female | 97.2 | -5.34 | 16.9 | 17.9 | 0.81 | 0.660 | 1 |
| MEC-04 | 8 | Male | 106.7 | -3.58 | 23.3 | 20.5 | 1.60 | 0.809 | 1 |
| MEC-05 | 6 | Male | 92.0 | -4.81 | 18.3 | 21.6 | 2.40 | 0.656 | 1 |
| MEC-06 | 9 | Female | 109.0 | -3.66 | 24.1 | 20.3 | 1.37 | 0.833 | 1 |
| Twice a day administration of meclizine | | | | | | | | | |
| MEC-07 | 7 | Male | 104.1 | -3.93 | 21.1 | 19.5 | 1.40 | 0.762 | 1 |
| MEC-08 | 7 | Male | 102.6 | -3.54 | 21.2 | 20.1 | 1.76 | 0.755 | 1 |
| MEC-09 | 10 | Female | 116.9 | -3.71 | 30.1 | 22.0 | 1.43 | 0.964 | 2 |
| MEC-10 | 10 | Male | 116.0 | -3.37 | 32.3 | 24.0 | 1.80 | 0.987 | 1 |
| MEC-11 | 5 | Female | 83.1 | -5.64 | 13.3 | 19.3 | 2.01 | 0.532 | 1 |
| MEC-12 | 5 | Female | 80.6 | -6.06 | 15.3 | 23.6 | 3.36 | 0.552 | 1 |

[a] Standard deviation

[b] Body mass index = Weight (kg) $\div$ Height (m)$^2$

[c] Body surface area = Height$^{0.725}$ × Weight$^{0.425}$ × 0.007184

Data was calculated using Excel-based Clinical Tools for Growth Evaluation of Children created by the Japanese Society for Pediatric Endocrinology (http://jspe.umin.jp/medical/chart_dl.html))

developed nausea, headache, and vomiting, and subject MEC-12 developed pyrexia. All AEs were mild and recovered without complications. The AEs seen in MEC-06 and MEC-12 were not considered to be related to the study drug, because the AEs confirmed at Day 4 in MEC-06 had occasionally appeared when she was in poor health, and laboratory tests showed increased inflammatory response prior to administration of meclizine in MEC-12. An AE that could not be denied a causal relationship with meclizine, therefore, was only somnolence observed on the day of administration in MEC-05. A 5-year-old girl (MEC-12), who had shown a slightly elevated WBC and CRP in blood screening just before medication, had a low-grade fever (pyrexia) 4 hours after administration of meclizine. She was dropped before the second administration of meclizine, because she received an acetaminophen tablet for relief of her symptoms, which was one of the prohibited concomitant drugs. MEC-12 was thus excluded from the PK analysis per protocol sets. With the exception of MEC-12, all procedures including the blood collection time were per protocol, and there was no deviation.

PK parameters were determined from 11 subjects except for one subject (MEC-12). The individual plasma concentration profiles of meclizine in ACH children after oral administration of "once a day group" and "twice a day group" are presented in Figs 2A and 3A, respectively. Body weight normalized PK profiles in each group are shown in Figs 2B and 3B. Meclizine was rapidly absorbed, being detectable in the plasma of all participants at one hour post-dose. The summary of PK parameters from 6 subjects of "once a day group" is shown in Table 2. The peak plasma concentration of meclizine was generally reached between 1 and 3 hours after oral administration, which resulted in the mean $C_{max}$ and $T_{max}$ of 130 ng/mL (range, 61.1–231 ng/mL), and 1.7 hours (range, 1–3 hours), respectively. Plasma concentration above the lower limit of quantification (0.5 ng/mL) was measured even after 7 days of administration in three out of 6 subjects (range, 0.705–3.23 ng/mL). The mean $AUC_{0-24h}$ was 761 ng·h/mL (range, 292–1650 ng·h/mL). The t1/2 during 24 hours were 8.5 hours (range, 6.9–11.1 hours). In "twice a day group", on the other hand, the $T_{max}$ was reached at an average of 2.6 hours (range, 1–4 hours) after the first dose, and the mean $C_{max}$ was 223 ng/mL (range, 118–474 mg/mL) (Table 3). In the second dose, the second $T_{max}$ reached at an average of 13.6 hours (range, 12–14 hours) after the first dose and the mean $C_{max}$ was 149 ng/mL (range, 100–276 ng/mL). All 5 subjects of "twice a day group" showed a measurable plasma concentration of meclizine (range, 0.734–4.88 ng/ml) 7 days after the first dose. The mean $AUC_{0-24h}$ was 2030 ng·h/mL (range, 1120–4670 ng·h/mL), and the t1/2 during 24 hours was 3.6 hours (range, 3.0–4.6 hours).

The simulation of repeated administration of meclizine for 14 days using the kel calculated based on the mean measured results (body weight normalized) after once a day administration of the drug demonstrated that plasma concentration apparently reached steady state around 10 days after the first dose both at once a day and twice a day administration (Figs 4 and 5). Similar results were obtained when simulated using the kel calculated based on the mean measured results after twice a day administration of meclizine (S1 Fig). We next performed simulation studies for specific two subjects (MEC-01 and MEC-02) who showed the most gradual disappearance of the drug from 24 hours to 7 days after completion of administration. Plasma concentration of MEC-01 and MEC-02 also reached steady state around 10 days and 12 days, respectively (S2 and S3 Figs).

Since meclizine was administered under fasting in "once a day group" and one hour after the controlled diet in "twice a day group", we next examined the influence of the diet on plasma concentration of meclizine. The $AUC_{0-10h}$ of the fasted and fed condition determined from the mean plasma concentration of each group normalized by body weight were 504 ng·h/mL and 813 ng·h/mL, respectively (Fig 6). Exposure of meclizine increased 1.6 times with the diet.

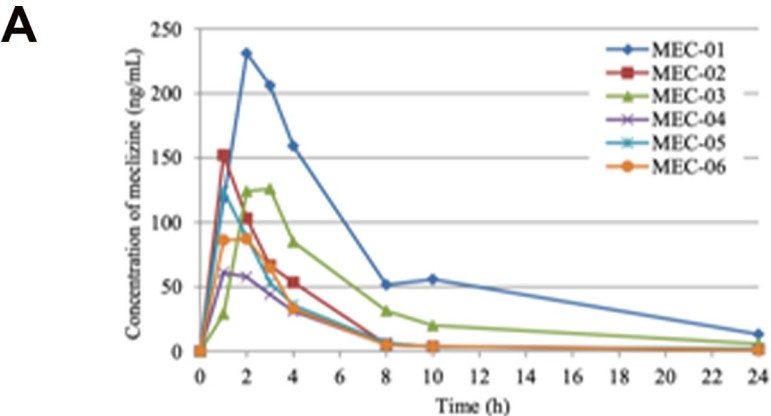

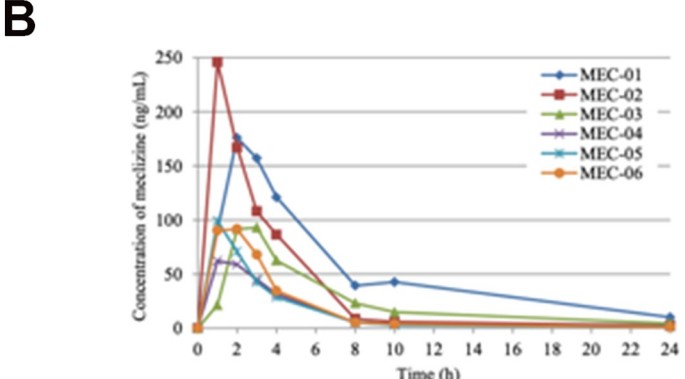

**Fig 2. Plasma concentration of meclizine from pre-dosing to 24 hours after single oral administration of meclizine hydrochloride 25 mg tablet.** Individual plasma concentration profiles of meclizine in ACH children (A) and body weight normalized profiles by the mean body weight of MEC-01 to MEC-06 (22.9 kg) (B).

## Discussion

This phase Ia, first-in-human study under the GCP and the current regulatory requirements evaluated the safety, tolerability, and PK parameters of meclizine administered once a day or twice a day in each 6 ACH children aged from 5 to 10 years. The PK data indicated that meclizine was rapidly absorbed following a single oral dose of 25 mg, with a mean $T_{max}$ of 1.7 hours and $C_{max}$ of 130 ng/mL in the fasting condition. After reaching $C_{max}$, the plasma concentration of meclizine decreased, with a mean t1/2 of 8.5 hours, and the mean $AUC_{0-24h}$ was 671 ng·h/mL. Previous PK parameters of meclizine administered to adult whose mean age of 26.7 years and mean body weight of 70.6 kg indicated that the mean values of $C_{max}$, $T_{max}$, $AUC_{0-24h}$, and t1/2 were 80.07 mg/mL, 3.11 hours, and 544.29 ng·h/mL and 5.21 hours, respectively [16]. Similar PK but higher drug exposure which probably result from smaller body weight, was confirmed in ACH children. Despite larger $C_{max}$ and $AUC_{0-24h}$, a single administration of meclizine was safe and well tolerated with no serious AEs in the current study.

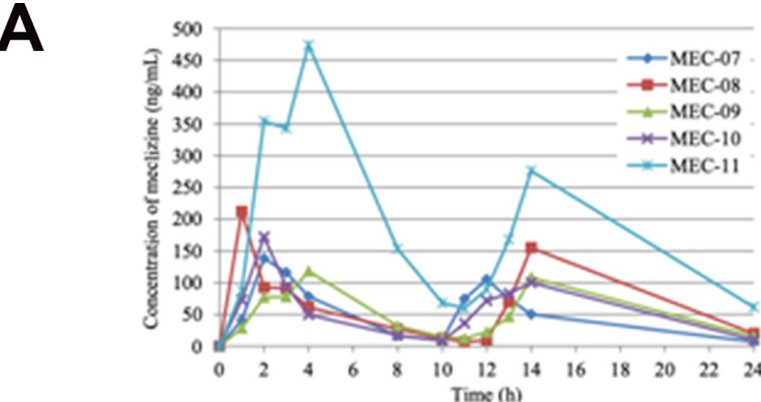

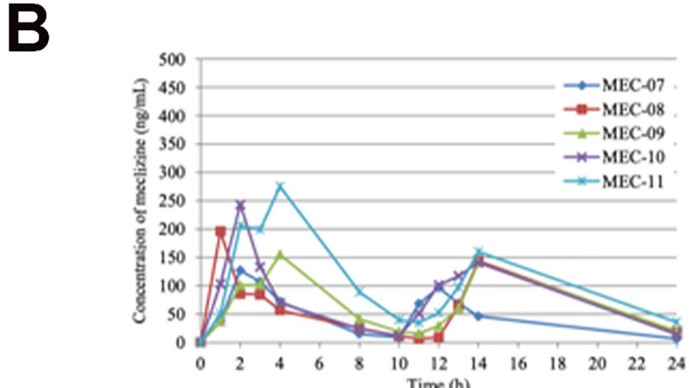

**Fig 3. Plasma concentration of meclizine from pre-dosing to 24 hours after twice a day oral administration of meclizine hydrochloride 25 mg tablet.** Individual plasma concentration profiles of meclizine in ACH children (A) and body weight normalized profiles by the mean body weight of MEC-01 to MEC-06 (22.9 kg) (B).

Some subjects (MEC-01 and MEC-11) showed higher concentration of meclizine than other subjects in both fed and fasted situations. The difference in age, gender, sexual maturity status, height, degree of obesity might cause differences in concentration of meclizine among the subjects, but it is difficult to draw conclusions because the sample size is too small. CYP2D6 was found to be the major enzyme for metabolism of meclizine, and its genetic polymorphism could contribute to the relatively large interindividual variability [16].

Repeated administration of meclizine during growing period will be required for the treatment of short stature in ACH. Some subjects showed above the lower limit of plasma meclizine concentration 7 days after administration, however, the simulation results indicated little accumulation for repeated administration. We additionally simulated the PK of meclizine after the 14th day administration using the elimination rate constant between 10 and 24 hours in "once a day group" and 14 and 24 hours in "twice a day group" in all subjects (S4 Fig). Coefficient of variation (CV) for $C_{max}$ of meclizine among the both subject groups were 54% and 31%, respectively (S1 Table), which was reasonable for human PK analysis. These findings would be valuable for further development of meclizine in near future.

**Table 2. Plasma concentrations and pharmacokinetic parameters of meclizine in achondroplasia children after single oral administration of meclizine hydrochloride 25 mg tablet.**

| Subject ID No. | Plasma concentration (ng/mL) | | | | | | | | | $C_{max}$ | $T_{max}$ | $AUC_{0-24\,h}$ [a] | $t_{1/2}$ [b] |
|---|---|---|---|---|---|---|---|---|---|---|---|---|---|
| | Time post-dosing | | | | | | | | | (ng/mL) | (h) | (ng·h/mL) | (h) |
| | Pre | 1 h | 2 h | 3 h | 4 h | 8 h | 10 h | 24 h | 7 d | | | | |
| MEC-01 | BLQ [1] | 118 [1] | 231 [3][c] | 206 [3][c] | 159 [1] | 51.5 [1] | 55.7 [1] | 13.2 [1] | 3.23 [1] | 231 | 2 | 1650 | 7.6 |
| MEC-02 | BLQ [1] | 152 [1] | 103 [1] | 66.4 [1] | 53.4 [1] | 5.31 [1] | 3.71 [1] | 1.65 [1] | 0.705 [1] | 152 | 1 | 512 | 10.3 |
| MEC-03 | BLQ [1] | 29.0 [1] | 124 [1] | 126 [1] | 84.8 [1] | 31.3 [1] | 20.0 [1] | 5.84 [1] | 0.902 [1] | 126 | 3 | 786 | 7.0 |
| MEC-04 | BLQ [1] | 61.1 [3] | 57.6 [3] | 44.1 [3] | 31.0 [1] | 5.10 [1] | 3.52 [1] | 1.18 [1] | BLQ [1] | 61.1 | 1 | 292 | 8.0 |
| MEC-05 | BLQ [1] | 124 [3] | 88.5 [3] | 53.2 [3] | 35.8 [1] | 6.59 [1] | 3.39 [1] | 1.95 [1] | BLQ [1] | 124 | 1 | 416 | 11.1 |
| MEC-06 | BLQ [1] | 86.1 [3] | 87.1 [3] | 64.8 [3] | 32.9 [1] | 4.97 [1] | 3.76 [1] | 0.964 [1] | BLQ [1] | 87.1 | 2 | 372 | 6.9 |
| Mean | BLQ | 95.0 | 115 | 93.4 | 66.2 | 17.5 | 15.0 | 4.13 | 0.806 | 130 | 1.7 | 671 | 8.5 |
| SD | NC | 45.2 | 61 | 62.2 | 49.8 | 19.6 | 21.0 | 4.79 | 1.253 | 59 | 0.8 | 509 | 1.8 |
| N | 6 | 6 | 6 | 6 | 6 | 6 | 6 | 6 | 6 | 6 | 6 | 6 | 6 |

[]: Dilution factor

BLQ: Below the lower limit of quantification (<0.5 ng/mL)

SD: standard deviation

NC: Not calculated

[a] The values of 0 hour were calculated using the predosing (Pre) values.

[b] $t_{1/2}$ values were calculated until 24 hours after the first dosing.

[c] Since the measured values exceeded the upper limit of the calibration curve, the samples were diluted and reanalyzed. The reanalysis data were employed.

The findings of the food effect demonstrated that absorption of meclizine was slightly delayed but overall exposure increased with diet. The effect of food on drug bioavailability can be mediated via a number of mechanisms including gastric emptying rate and a change in gastrointestinal pH, which can significantly affect pharmacological response or drug safety [17]. Delayed absorption and increased exposure of meclizine when administered after a meal could be attributable to food-induced delay in gastric emptying rate and a high fat solubility of meclizine. The difference of meclizine absorption, however, was not considered serious in fed and fasted states, and increased exposure of the drug was not considered a clinically relevant issue

**Table 3. Pharmacokinetic parameters of meclizine in achondroplasia children after twice a day oral administration of meclizine hydrochloride 25 mg tablet.**

| Subject ID No. | First dosing | | Second dosing | | | |
|---|---|---|---|---|---|---|
| | $C_{max}$ | $T_{max}$ | $C_{max}$ | $T_{max}$ [a] | $AUC_{0-24h}$ [b] | $t_{1/2}$ [c] |
| | (ng/mL) | (h) | (ng/mL) | (h) | (ng·h/mL) | (h) |
| MEC-07 | 138 | 2 | 105 | 12 | 1120 | 3.3 |
| MEC-08 | 212 | 1 | 155 | 14 | 1690 | 3.4 |
| MEC-09 | 118 | 4 | 108 | 14 | 1350 | 3.6 |
| MEC-10 | 172 | 2 | 100 | 14 | 1320 | 3.0 |
| MEC-11 | 474 | 4 | 276 | 14 | 4670 | 4.6 |
| Mean | 223 | 2.6 | 149 | 13.6 | 2030 | 3.6 |
| SD | 145 | 1.3 | 74 | 0.9 | 1490 | 0.6 |
| N | 5 | 5 | 5 | 5 | 5 | 5 |

SD: standard deviation

[a] $T_{max}$ values for the second dosing were indicated as the time from the first dosing.

[b] The values of 0 hour were calculated using the predosing (Pre) values.

[c] $t_{1/2}$ values were calculated until 24 hours after the first dosing.

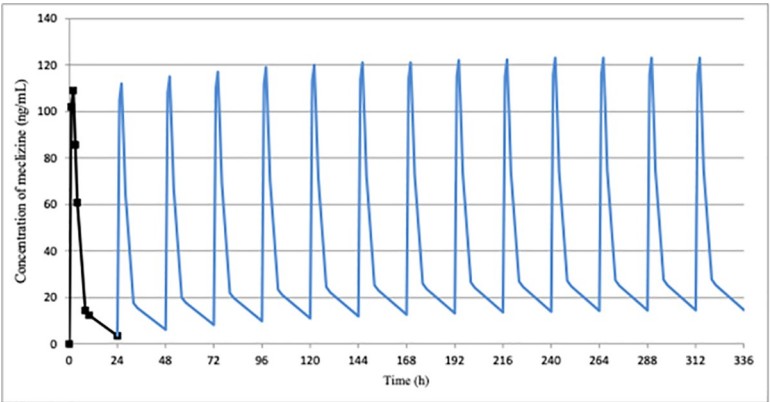

**Fig 4. Simulated plasma concentration profile of meclizine at once a day for 14 days multiple administrations in ACH children.** Plasma concentration simulated using the mean measured results after once a day administration of meclizine hydrochloride 25 mg tablet (body weight normalized) apparently reached steady state around 10 days after the first dose.

when meclizine was administered after food. Therefore, it is recommended that meclizine be taken either fed or fasted condition in further trials.

An optimal drug dosage is based on both the PK and the patient's specific characteristics which may eventually affect drug metabolism, and this should be taken into consideration to minimize toxicity and prevent subtherapeutic levels. ACH is characterized by not only disproportionate short stature but also early obesity which represents a major health problem in these patients, affecting approximately 50% of them during childhood [18]. The effects of obesity on drug dosage in the adult population are well documented, but the PK assessment of drugs used in children is more limited. Obese patients have a larger amount of fat body mass and lean body mass, and a higher proportion of extracellular water compared to total body water [19, 20]. These alterations may alter PK parameters and provide significant consequences on ACH children [21]. The PK and safety data obtained from the current phase Ia study, therefore, is valuable in the process of further clinical trials for the treatment of short stature in ACH children.

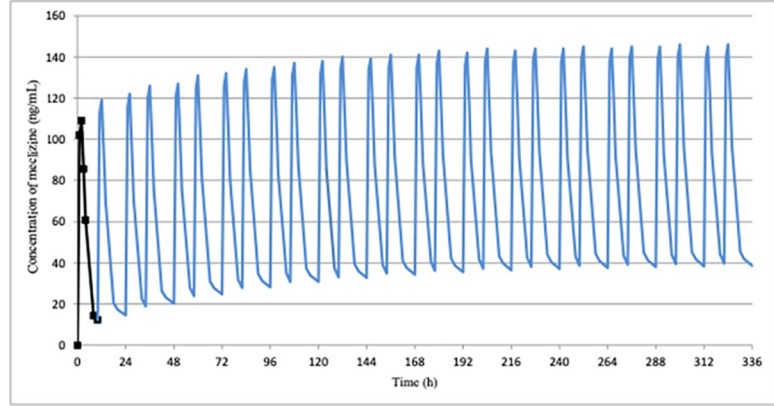

**Fig 5. Simulated plasma concentration profile of meclizine at twice a day for 14 days multiple administrations in ACH children.** Plasma concentration simulated using the mean measured results after once a day administration of meclizine hydrochloride 25 mg tablet (body weight normalized) also reached steady state around 10 days after the first dose.

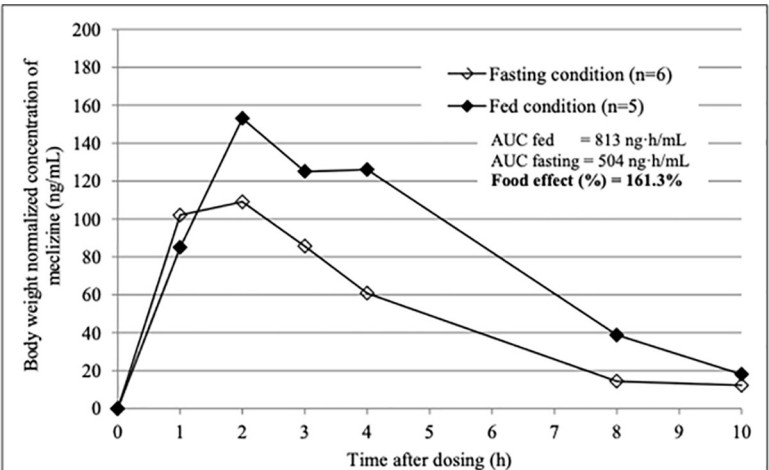

**Fig 6. Body weight normalized plasma concentration of meclizine in the fasting and fed condition.** The $AUC_{0-10h}$ of the fasted and fed condition were 504 ng·h/mL and 813 ng·h/mL, respectively.

## Conclusions

Meclizine was rapidly absorbed after oral administration and showed higher exposure in children than in adults, and in the fed condition than in the fasted condition. Simulation studies of repeated administration of meclizine for 14 days indicated that steady state was reached within 14 days at the latest. Oral administration of meclizine once a day or twice a day seemed to be safe and well tolerated with no serious adverse events in ACH children.

## Supporting information

**S1 Checklist. TREND statement checklist.**
(PDF)

**S2 Checklist.**
(PDF)

**S1 Fig. Simulated plasma concentration profile of meclizine at twice a day for 14 days multiple administrations in ACH children using the mean measured results after twice a day administration of meclizine hydrochloride 25 mg tablet (body weight normalized).** Plasma concentration apparently reached steady state around 10 days after the first dose.
(TIFF)

**S2 Fig.** Simulated plasma concentration profile of meclizine at once a day **(A)** and twice a day **(B)** for 14 days multiple administrations using the plasma concentration of MEC-01 after single administration of meclizine hydrochloride 25 mg tablet. Plasma concentration reached steady state around 10 days and 12 days after the first dose at once a day and twice a day multiple administrations, respectively.
(TIF)

**S3 Fig.** Simulated plasma concentration profile of meclizine at once a day **(A)** and twice a day **(B)** for 14 days multiple administrations using the plasma concentration of MEC-02 after single administration of meclizine hydrochloride 25 mg tablet. Plasma concentration reached steady state around 10 days after the first dose both at once and twice a day multiple

administrations.
(TIF)

**S4 Fig.** Simulated plasma concentration profile of meclizine after the 14th day administration using the elimination rate constant between 10 and 24 hours in "once a day group" **(A)** and 14 and 24 hours in "twice a day group" **(B)**.
(TIF)

**S1 Table. Simulated pharmacokinetic parameters of meclizine in achondroplasia children after the 14th day administration using the elimination rate constant between 10 and 24 hours in "once a day group" and 14 and 24 hours in "twice a day group".**
(PDF)

**S1 File.** Simulated plasma concentration profile of meclizine at once a day **(A)** and twice a day **(B)** for 14 days multiple administrations in each individual.
(TIF)

## Acknowledgments

The authors would like to thank the patients for their participation in this study. The authors acknowledge Yuko Sudo and Chika Namekata, Clinical Research Coordinator (CRC) of Nagoya University Hospital, for their contribution to this study as CRC, and Naoko Hayashi and Natsuko Tamura for their monitoring of this study. The authors also acknowledge Asako Ito for her secretarial assistance. All authors reviewed and approved the article submission.

## Author Contributions

**Conceptualization:** Kohei Ueda.

**Data curation:** Hiroshi Kitoh, Masaki Matsushita, Kenichi Mishima.

**Formal analysis:** Yachiyo Kuwatsuka, Hiroshi Morikawa, Yasuhiro Nakai.

**Funding acquisition:** Hiroshi Kitoh.

**Investigation:** Hiroshi Kitoh, Masaki Matsushita, Kenichi Mishima, Tadashi Nagata, Yasunari Kamiya.

**Methodology:** Yachiyo Kuwatsuka.

**Project administration:** Kohei Ueda.

**Supervision:** Naoki Ishiguro.

**Validation:** Hiroshi Morikawa.

**Writing – original draft:** Hiroshi Kitoh.

**Writing – review & editing:** Hiroshi Kitoh, Kohei Ueda, Yasuhiro Nakai.

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
