## [Decision Letter · Decision Letter 0]

28 Oct 2019

PONE-D-19-20712

Pharmacokinetics and safety after once and twice a day doses of meclizine hydrochloride administered to children with achondroplasia

PLOS ONE

Dear Dr Kitoh,

Thank you for submitting your manuscript to PLOS ONE. After careful consideration, we feel that it has merit but does not fully meet PLOS ONE’s publication criteria as it currently stands. Therefore, we invite you to submit a revised version of the manuscript that addresses the points raised during the review process.

The manuscript has been assessed by two reviewers, their comments are available below.

The reviewers have raised concerns that need attention in a revision. The reviewers feel that the manuscript should report additional information about the participants and report a justification for the sample size employed. The reviewers note that the statistical analyses should be reported in greater detail and raise that the differences in concentration between patients should be further considered and discussed.

Could you please revise the manuscript to address the items raised.

We would appreciate receiving your revised manuscript by Dec 10 2019 11:59PM. Please include the following items when submitting your revised manuscript:

We look forward to receiving your revised manuscript.

Kind regards,

Iratxe Puebla

Senior Managing Editor, PLOS ONE

Journal Requirements:

HK received Advanced Medical Development Funds of Nagoya University Hospital. The funders had no role in study design, data collection and analysis, decision to publish, or preparation of the manuscript.

We note that one or more of the authors are employed by a commercial company: Ina Research Inc.

Reviewers' comments:

Reviewer's Responses to Questions

**Comments to the Author**

1. Is the manuscript technically sound, and do the data support the conclusions?

Reviewer #1: Partly

Reviewer #2: No

2. Has the statistical analysis been performed appropriately and rigorously? 

Reviewer #1: Yes

Reviewer #2: No

3. Have the authors made all data underlying the findings in their manuscript fully available?

Reviewer #1: Yes

Reviewer #2: Yes

4. Is the manuscript presented in an intelligible fashion and written in standard English?

Reviewer #1: Yes

Reviewer #2: Yes

5. Review Comments to the Author

Reviewer #1: Authors analyzed pharmacokinetics of meclizine hydrochloride administration to achondroplasia patient.

Major comments:

1. Some patients to whom administrated meclizine hydrochloride shows higher concentration than other patients in both of fed and non-fed situations (MEC-01 and MEC-11). Although authors also showed body weight normalized concentration, some patient (MEC-01, 02, 08,10,11) still showed higher concentration compared to other patient. Reviewer wonders why these difference in concentration among the patients occurred and is afraid that these difference might cause the difference of growth promoting effects among patients. Reviewer speculates that the difference in age, sex, sexual maturity status (especially in female because puberty comes earlier in female than in male), height, degree of obesity might cause difference of concentration. In this meaning, age, sex, height SD score, BMI and BMI-SD score, body surface area and Tanner stage of each patient should be shown and why these difference in concentration occurred should be discussed.

2. Reviewer also wonders how the concentration in each patient during 14 days administration was. Is there any difference in concentration among the patient? Please show the pharmacokinetics of each patient during 14 days administration.

Reviewer #2: A phase 1a study was conducted to evaluate PK and the safety of meclizine when administered to 12 ACH children. The drug was well tolerated in ACH children with no serious adverse events.

Major Revisions:

1- Line 124: Provide details of how the simulation was conducted with enough granularity that would allow one to replicate the method.

2- Line 153: Mean is the standard terminology for average. Provide a measure of the distribution of the ages, weights and height. Consider including or substituting body mass index for weight and height. Since the sample size is small summarizing the data with medians and interquartile ranges may be more appropriate.

3- Include the standard table 1 containing a summary of subject baseline characteristics separated by once or twice a day drug administration schedule.

Minor Revisions:

1- Line 119: Site a reference for calculating the PK AUC.

2- Lines 141 and 143: Consider changing "population" to "analysis."

3- Line 141: Indicate the statistical approach used for calculating the 95% confidence intervals.

4- Line 151: Typographical error: signed

5- Cite the statistical software used for the analysis.

6- Line 187-8 and 190 and subsequently: Indicate the summary values included inside the parenthesis.

7- Line 188: Provide a measure of dispersion for the time.

8- Define all abbreviations in table 2. Standard deviation is typically SD.

9- For better clarity, on each figure label the y-axis with an indication of the the concentration of meclizine and the time-frame.

10- State and provide justification for the study's target sample size.

6. PLOS authors have the option to publish the peer review history of their article (what does this mean?). If published, this will include your full peer review and any attached files.

Reviewer #1: No

Reviewer #2: No

---

## [Author Response · Author response to Decision Letter 0]

6 Nov 2019

We revised our manuscript according to the comments of the reviewers. Attached please find a Response to Reviewers file. The sentences and words we changed are written in red in the revised manuscript with track changes. Line numbers noted in the file refer to the revised manuscript. We changed Figures 2-6, Supporting Figures 1-3, and Table 2 and 3 (original Table 1 and 2). We created Table 1, Supporting Figure 4, Supporting File 1, and Supporting Table 1. We added “Reference 15”.

---

## [Decision Letter · Decision Letter 1]

6 Feb 2020

PONE-D-19-20712R1

Pharmacokinetics and safety after once and twice a day doses of meclizine hydrochloride administered to children with achondroplasia

PLOS ONE

Dear Dr Kitoh,

Thank you for submitting your manuscript to PLOS ONE. After careful consideration, we feel that it has merit but does not fully meet yet PLOS ONE’s publication criteria as it currently stands. Therefore, we invite you to submit a revised version of the manuscript that addresses the points raised during the review process and the subsequent assessment by the editor. 

We would appreciate receiving your revised manuscript by Mar 22 2020 11:59PM. To enhance the reproducibility of your results, we recommend that if applicable you deposit your laboratory protocols in protocols.io, where a protocol can be assigned its own identifier (DOI) such that it can be cited independently in the future. For instructions see: http://journals.plos.org/plosone/s/submission-guidelines#loc-laboratory-protocols

We look forward to receiving your revised manuscript.

Kind regards,

Karel Allegaert

Academic Editor

PLOS ONE

Journal Requirements:

Additional Editor Comments :

the general statement on safety and tolerance in the abstract is based on a very low number of cases as phase 1 studies are not designed to assess safety, to that the editor suggest to adapt this sentence somewhat in the abstract

Reviewers' comments:

Reviewer's Responses to Questions

**Comments to the Author**

1. If the authors have adequately addressed your comments raised in a previous round of review and you feel that this manuscript is now acceptable for publication, you may indicate that here to bypass the “Comments to the Author” section, enter your conflict of interest statement in the “Confidential to Editor” section, and submit your "Accept" recommendation.

Reviewer #1: (No Response)

Reviewer #2: All comments have been addressed

2. Is the manuscript technically sound, and do the data support the conclusions?

Reviewer #1: Yes

Reviewer #2: (No Response)

3. Has the statistical analysis been performed appropriately and rigorously? 

Reviewer #1: Yes

Reviewer #2: (No Response)

4. Have the authors made all data underlying the findings in their manuscript fully available?

Reviewer #1: Yes

Reviewer #2: (No Response)

5. Is the manuscript presented in an intelligible fashion and written in standard English?

Reviewer #1: Yes

Reviewer #2: (No Response)

6. Review Comments to the Author

Reviewer #1: The manuscript is well revised according to the reviewer’s suggestion and critique except for the addition of BMI-SD score to the Table 1.

Average of BMI during child period differs according to the age and sex in contrast to the adult whose standard level of BMI is same independent of age and sex.

So, it is not appropriate to show the median value among patients of different age and sex in result section (line 165〜166). The BMI-SD score of each patient should be calculated by using the growth evaluation file for children created by the Japanese society of pediatric endocrinology (available at http://jspe.umin.jp/medical/chart_dl.html) and add the each value in Table 1.

Reviewer #2: (No Response)

7. PLOS authors have the option to publish the peer review history of their article (what does this mean?). If published, this will include your full peer review and any attached files.

Reviewer #1: No

Reviewer #2: No

---

## [Author Response · Author response to Decision Letter 1]

7 Feb 2020

Additional Editor Comments)

The general statement on safety and tolerance in the abstract is based on a very low number of cases as phase 1 studies are not designed to assess safety, to that the editor suggest to adapt this sentence somewhat in the abstract.

Response)

Agree with the editor’s comment, we revised the last sentence of abstract and conclusions. 

Correction)

Line 17-19:

Although higher drug exposure was confirmed in ACH children compared to adults, a single administration of meclizine seemed to be well tolerated. 

Line 317-319:

Oral administration of meclizine once a day or twice a day seemed to be safe and well tolerated with no serious adverse events in ACH children.

Comments from reviewer 1)

The manuscript is well revised according to the reviewer’s suggestion and critique except for the addition of BMI-SD score to the Table 1. Average of BMI during child period differs according to the age and sex in contrast to the adult whose standard level of BMI is same independent of age and sex. So, it is not appropriate to show the median value among patients of different age and sex in result section (line 165〜166). The BMI-SD score of each patient should be calculated by using the growth evaluation file for children created by the Japanese society of pediatric endocrinology 

(available at http://jspe.umin.jp/medical/chart_dl.html) and add the each value in Table 1.

Response)

Thank you very much for your valuable comments. I totally agree with you in regard to children’s BMI. We re-calculated data from subject baseline characteristics using Excel-based Clinical Tools for Growth Evaluation of Children created by the Japanese Society of Pediatric Endocrinology. We added the BMD-SDS in Table 1. We also deleted the sentence regarding median value of BMI, and instead described the BMI-SDS in the text. 

Correction)

Line 164-166: 

The body mass index-standard deviation score (BMI-SDS) ranged from 0.81 to 2.40 in “once a day group” and from 1.40 to 3.36 in “twice a day group”.

Line 176-177: Table 1 was re-created.

---

## [Editor Report · Decision Letter 2]

12 Feb 2020

Pharmacokinetics and safety after once and twice a day doses of meclizine hydrochloride administered to children with achondroplasia

PONE-D-19-20712R2

Dear Dr. Kitoh,

We are pleased to inform you that your manuscript has been judged scientifically suitable for publication and will be formally accepted for publication once it complies with all outstanding technical requirements.

With kind regards,

Karel Allegaert

Academic Editor

PLOS ONE

Additional Editor Comments (optional):

thank you for your additional minor adaptations.
---

## [Editor Report · Acceptance letter]

14 Feb 2020

PONE-D-19-20712R2 

Pharmacokinetics and safety after once and twice a day doses of meclizine hydrochloride administered to children with achondroplasia 

Dear Dr. Kitoh:

I am pleased to inform you that your manuscript has been deemed suitable for publication in PLOS ONE. Congratulations! Your manuscript is now with our production department. 

With kind regards,

on behalf of

Dr. Karel Allegaert 

Academic Editor

PLOS ONE